# Remote Sensing Monitoring of Pine Wilt Disease Based on Time-Series Remote Sensing Index

Lin Long [1], Yuanyuan Chen [1], Shaojun Song [2], Xiaoli Zhang [1,*], Xiang Jia [1], Yagang Lu [3] and Gao Liu [4]

1 Beijing Key Laboratory of Precision Forestry, Forestry College, Beijing Forestry University, Beijing 100083, China
2 School of Foreign Languages, Beijing Forestry University, Beijing 100083, China
3 East China Institute of Investigation and Planning, National Forestry and Grassland Administration, Hangzhou 310019, China
4 Lu'an Forestry Bureau of Anhui Province, Lu'an 237000, China
* Correspondence: zhangxl@bjfu.edu.cn; Tel.: +86-010-6233-6227

**Abstract:** Under the strong influence of climate change and human activities, the frequency and intensity of disturbance events in the forest ecosystem both show significant increasing trends. Pine wood nematode (*Bursapherenchus xylophilus*, PWN) is one of the major alien invasive species in China, which has rapidly infected the forest and spread. In recent years, its tendency has been to spread from south to north, causing serious losses to Pinus and non-Pinus coniferous forests. It is urgent to carry out remote sensing monitoring and prediction of pine wilt disease (PWD). Taking Anhui Province as the study area, we applied ground survey, satellite-borne optical remote sensing imagery and environmental factor statistics, relying on the Google Earth Engine (GEE) platform to build a new vegetation index NDFI based on time-series Landsat images to extract coniferous forest information and used a random forest classification algorithm to build a monitoring model of the PWD infection stage. The results show that the proposed NDFI differentiation threshold classification method can accurately extract the coniferous forest range, with the overall accuracy of 87.75%. The overall accuracy of the PWD monitoring model based on random forest classification reaches 81.67%, and the kappa coefficient is 0.622. High temperature and low humidity are conducive to the survival of PWN, which aggravates the occurrence of PWD. Under the background of global warming, the degree of PWD in Anhui Province has gradually increased, and has transferred from the southwest and south to the middle and northeast. Our results show that PWD monitoring and prediction at a regional scale can be realized by using long time-series multi-source remote sensing data, NDFI index can accurately extract coniferous forest information and grasp disease information in a timely manner, which is crucial for effective monitoring and control of PWD.

**Keywords:** pine wood nematode; Google Earth Engine; time-series; vegetation index; remote sensing monitoring





## 1. Introduction

Pine wood nematode (*Bursapherenchus xylophilus*, PWN) is a kind of forest pest that spreads through the insect vector *Monochamus alternatus* in a specific environment and invades the Pinus Linn plant, which is one of the major alien invasive species in China [1], causing rapid wilt and death of pine trees [2,3]. PWN was first found in North America. It was introduced into Nagasaki, Japan, in 1905, and then gradually spread to Europe and Asian countries. It was not until 1971 that PWN was confirmed as a pathogen. It is listed as a quarantine pest in more than 40 countries around the world [4]. Pine wilt disease (PWD) is a fatal disease caused by invasion and parasitism of PWN, known as the 'cancer of pine trees' [5], also referred as the COVID-19 virus of pinewoods. It only takes about 40 days from the invasion of PWN to reducing the resin secretion to its death, and the forest

will be destroyed in 3–5 years. The high spread-speed of PWD leads to great difficulty controlling it.

Due to the suitable conditions for the survival of PWN in China, PWD was first found in Nanjing in 1982. In the past four decades, the disease has spread to the middle temperate zone, showing a rapid westward and northward spread. It has spread to 5479 township administrative regions in 742 county-level administrative regions of 19 provinces in China, covering an area of 1.81 million hectares [6], posing a huge threat to the balance of pine forest resources and ecosystems in China. It has invaded many national scenic spots and key ecological areas, which not only affects the development of the social economy, but also seriously damages the natural landscape and ecological environment [7,8]. Accurate prevention and control of this disease is of top priority in current forest protection.

Anhui Province belongs to the transitional area of warm temperate zone and subtropical zone. Because of its monsoon climate and four distinct seasons, the precipitation has obvious seasonal changes. It has diverse landforms including plains, platforms, hills, mountains, etc. Anhui Province is an important constituent province of the Yangtze River Delta Economic Circle, promoting the coordinated development pattern of 'one circle and five regions'. At the same time, as one of the provinces with rich pine forest resources in China [9], PWD has caused serious losses to the forest resources of this province since PWD was first found in 1988, so monitoring and prevention of PWD in Anhui Province is particularly important, and has indicative and representative significance for PWD disaster prevention and control in other regions with the similar forest structure, climate characteristics and biogeographical environment.

Accurate and efficient monitoring is the premise of disease prevention and control. Sexual attraction and field survey are two traditional pest monitoring methods [10]. However, field survey is limited by climate, terrain and other conditions, with poor real-time performance, high cost, and low efficiency. It is unable to conduct macro dynamic monitoring, and cannot essentially identify the mechanism that the ecosystem structure is infected by PWD. Although the government has been making great efforts to coordinate the prevention and control of the disease, the traditional field investigation and monitoring methods often delays the best time for disease control, thus causing incalculable losses [11].

Remote sensing technology has incomparable advantages over conventional ground survey methods in forest pest monitoring research [12]. When the host plant of PWN is infected, the color of its needles will change from green to grayish green and yellow, until the whole crown becomes reddish brown; the plant will die, but the needles of the plant will not fall off [13]. This feature of needle discoloration of infected plants will produce a spectral feature change response on satellite-borne optical remote sensing imagery. Satellite-based remote sensing imagery has the advantage of a wide covering range and detectability of fine features of ground surfaces, which can be monitored in a large range and in real time. Therefore, remote sensing technology plays an incomparably important role in the study of forestry pests and diseases with its advantages of wide monitoring range, detectability of fine features of ground surfaces and so on. Because the vegetation has a special spectral characteristic curve, the most significant change of the host plant after infection by PWN is the variation of its external morphology and internal organizational structure of the leaves, which makes trees respond to spectral changes in remote sensing imagery [14]. Qin Lin et al. [15] used Beijing-3 data to conduct remote sensing monitoring of PWD through image fusion, index calculation and information extraction. The results showed that the accuracy and recall rate of intelligent extraction of discolored pine trees based on deep learning were high, which improved the solubility of quantitative monitoring of individual discolored pine tree, and was conducive to accurate monitoring, prevention and control of PWD. Qin Jun [16] used SCANet to obtain the spatial information of PWD, in order to enhance the spatial and spectral detailed features of the disease and integrate its shallow features. The results showed that the recall rate of PWD extracted by SCANet was 0.9322, which realized the rapid, high-precision and automated intelligent monitoring of PWD by unmanned aerial vehicle remote sensing. With the support of geostatistics,

machine learning and other technologies, it is possible to use image processing, information extraction, and parameter inversion for remote sensing monitoring of pests and diseases.

At present, satellite remote sensing monitoring also has many limitations, such as low recognition rate due to the limitation of spatial spectrum resolution, the difference between PWD-induced needle leaves discoloration and normal physiological discoloration of broad leaves, etc. [17]. In view of this, this study takes Anhui Province as the study area, and with the support of ground survey, satellite-borne optical remote sensing, geospatial analysis technology and machine learning technology, integrates the biological and ecological characteristics, living environment, human activities, spatial distribution of host plants and other factors of PWN, and uses the vegetation index of time-series to build disease degree monitoring model. Through the screening research on the influencing factors of disease occurrence, the meteorological change of long time-series and the analysis of disease occurrence and disease degree, reliable basis is provided for disease prevention and control. The focus and innovations of this research are as follows:

a    Proposing a new index for extracting the coniferous forest range based on time-series Landsat images.

b    Building a monitoring model of infection areas of PWD based on the time-series Landsat images.

c    Analyzing the spatio-temporal dynamics of PWD to strengthen the understanding of disaster occurrence and spread.

## 2. Materials and Methods

### 2.1. Study Area

Anhui Province is located in East China, with geographical coordinates ranging from 29°41′ to 34°38′N, 114°54′ to 119°37′E. It borders Shandong Province in the north, Jiangsu Province in the east, Jiangxi Province in the south, and Hubei Province in the west, with a total area of 140.1 thousand km². The landform of its territory includes mountains, hills, plains, etc. The mountains are mostly distributed in the northeast and extends roughly in an east–west direction. The mean annual precipitation is 773~1670 mm, and the mean annual temperature is 14~17 °C. Anhui Province is a key province of the southern collective forest area in China, with rich forest resources. According to the ninth national forest resources inventory statistics, the area of coniferous forests in Anhui Province accounts for 27.29% of the total forest area [18]. Among the coniferous forests, the forest area dominated by *Pinus massoniana* is the largest, followed by *Cunninghamia lanceolata*. According to the survey announcement of Anhui Forestry Bureau, Anhui Province strictly enforced quarantine and law enforcement on PWD, paid close attention to the source of infected trees, practically eliminated the hidden danger of epidemic transmission, and prevented external invasion and internal diffusion. The province has carried out a special campaign to control the violations of laws and regulations against infected trees, actively applied the refined supervision platform of the National Forestry and Grass Administration for PWD epidemics, and targeted the epidemic monitoring and control to the pine forest sub-compartments. At present, 14,599 sub-compartments of the PWD epidemic have been recorded, accounting for 95.50% of the total sub-compartments, providing more accurate data support for the prevention and control of the PWD epidemic.

We conducted a field survey in Huoshan County, Lu'an City, Anhui Province. Huoshan County is located at the northern foot of Ta-pieh Mountains, with typical mountain features. It is one of the key mountain counties in Anhui Province, with a mean annual temperature of 15.2 °C, the highest temperature of nearly 40 °C, and the lowest temperature of minus 10 °C. In 2014, sporadic dead pine trees suspected of PWD were found in Mozitan Town for the first time. Later, they were also found in other towns, and were identified as PWD at the end of the year [19]. In recent years, PWD has broken out seriously, and Huoshan County is one of the worst hit PWD areas in Anhui Province. The location and sample plot data of the study area are shown in Figure 1.

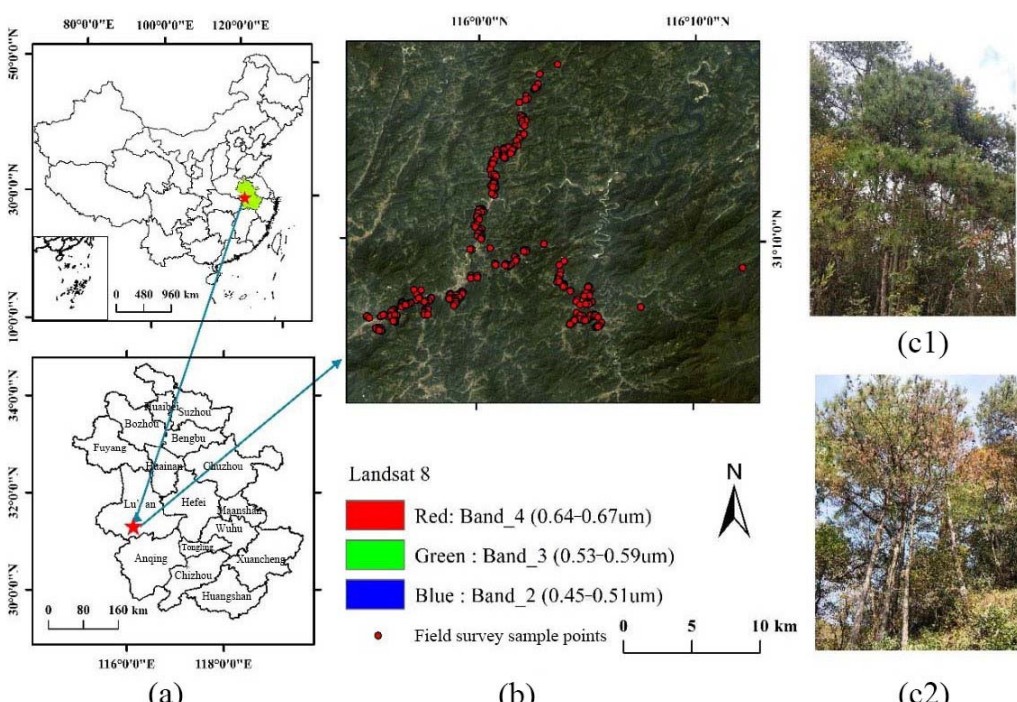

**Figure 1.** Location of study area and sample plot. (**a**) Location of the study area. (**b**) True color Landsat image and sample plot. (**c1**) Picture of healthy pine trees in the study area. (**c2**) Picture of infected pine trees in the study area.

*2.2. Data*

2.2.1. Field Survey Data

In order to obtain the damage characteristics of the different infection stages of the forest by PWD for extracting the diagnostic spectrum band, vegetation index, and texture information from the remotely sensed imagery, in October 2021, field research was conducted in Huoshan County, Lu'an City, Anhui Province to determine the location of the study area, and a field survey was conducted at the end of October of the same year. In the study area, randomly distributed single trees with an interval of more than 30m were selected to obtain the data of the infected single trees. RTK was used to obtain the location of the single tree infected by PWD. We recorded the detailed information of infected trees, and took photos of infected trees and the surrounding environment. A total of 614 single trees were obtained, including 249 infected single trees and 365 healthy single trees, in which the selection of healthy single trees met the requirements that the host plants within 30m around them were healthy.

2.2.2. Remote Sensing Imagery

The Landsat remote sensing data selected for this study was Landsat 8 OLI surface reflectance data, and the image data ID was 'LANDSAT/LC08/C01/T1_SR'. Considering that the onset time of the infected host plants was from late August to early November and the influence of cloud cover in the study area, as the single and cloudless Landsat images could not cover the entire province, the mean value data of the monthly cloud removal synthetic images from 2019 to 2020 was selected as the basic data source to extract the coniferous forest range, and the mean data of the cloud removal synthetic images from August to November of each year from 2018 to 2021 was selected as the basic data for monitoring the occurrence of PWD.

2.2.3. Other Auxiliary Data

The forest screened from the GlobeLand30 Version 2020 land cover data was selected as the basic data for extracting the coniferous forest in Anhui Province. With the spatial

resolution of 30m, the overall classification accuracy of 85.72%, and the kappa coefficient of 0.82.

Digital Elevation Model (DEM) data was from Shuttle Radar Topography Mission (SRTM) products with a spatial resolution of 30m released by the data center of United States Geological Survey (USGS, https://www.usgs.gov/products/data (accessed on 18 July 2021)).

All data were called on Google Earth Engine (GEE); GEE is a tool subordinate to Big Google, it can quickly batch process a large number of 'huge' satellite images, quickly calculate various vegetation indices, predict crop related yields, monitor drought growth, and monitor global forest changes. Its interface is intuitive and friendly to new users.

In this study, the statistical data of sub-compartment dominated by pine tree species in the Forest Management Inventory of Anhui Province in 2020 and the data of PWD epidemic area in Anhui Province over the years were obtained, which were provided by the forestry department of Anhui Province.

### 2.3. Method

This study takes Anhui Province as the study area, and with the support of ground survey, satellite-borne optical remote sensing, geospatial analysis technology and machine learning technology, integrates the biological and ecological characteristics, living environment, human activities, spatial distribution of host plants and other factors of PWN, and uses the vegetation index of time-series to build a disease degree monitoring model. Through the screening research on the influencing factors of disease occurrence, the meteorological change of long time-series and the analysis of disease occurrence and disease degree, a reliable basis is provided for disease prevention and control; its technical route is shown in Figure 2.

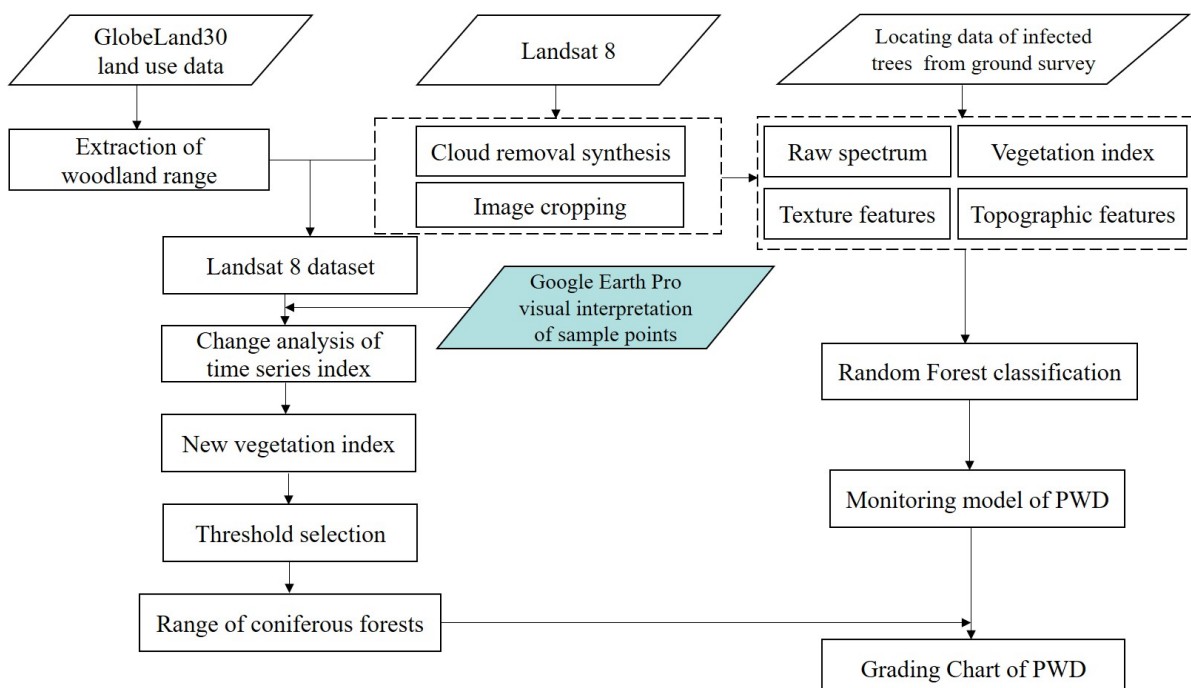

**Figure 2.** Technical route chart of this study.

### 2.4. Coniferous Forest Extraction Based on Time-Series Landsat Images

With the increasing adaptability of PWN in China in recent years, the host plants of PWN have gradually expanded from dozens of pine species such as *Pinus massoniana* and *Pinus densiflora* to other non-Pinus coniferous species [20]. According to the statistical data of the sub-compartment area in the forest resources inventory of Anhui Province, the

sub-compartment area with pine forest as the dominant tree species accounts for 25.09% of the forest area in Anhui Province. According to the ninth national forest resource inventory statistics in China, the coniferous forest area in Anhui Province accounts for 27.29% of the total forest area in the province. Therefore, this paper took the distribution area of coniferous forest in the study area as the potential adaptive area for PWN to conduct PWD monitoring and disease analysis.

### 2.4.1. Data Analysis and Selection

GlobeLand30 land cover data, as a product with high classification accuracy among the global same-type data, has been cited by many organizations [21]. In this study, the forests in the primary type of GlobeLand30 land cover data were used as the basis for information coniferous forest extraction. The range of forests in Anhui Province in 2020 is shown in Figure 3.

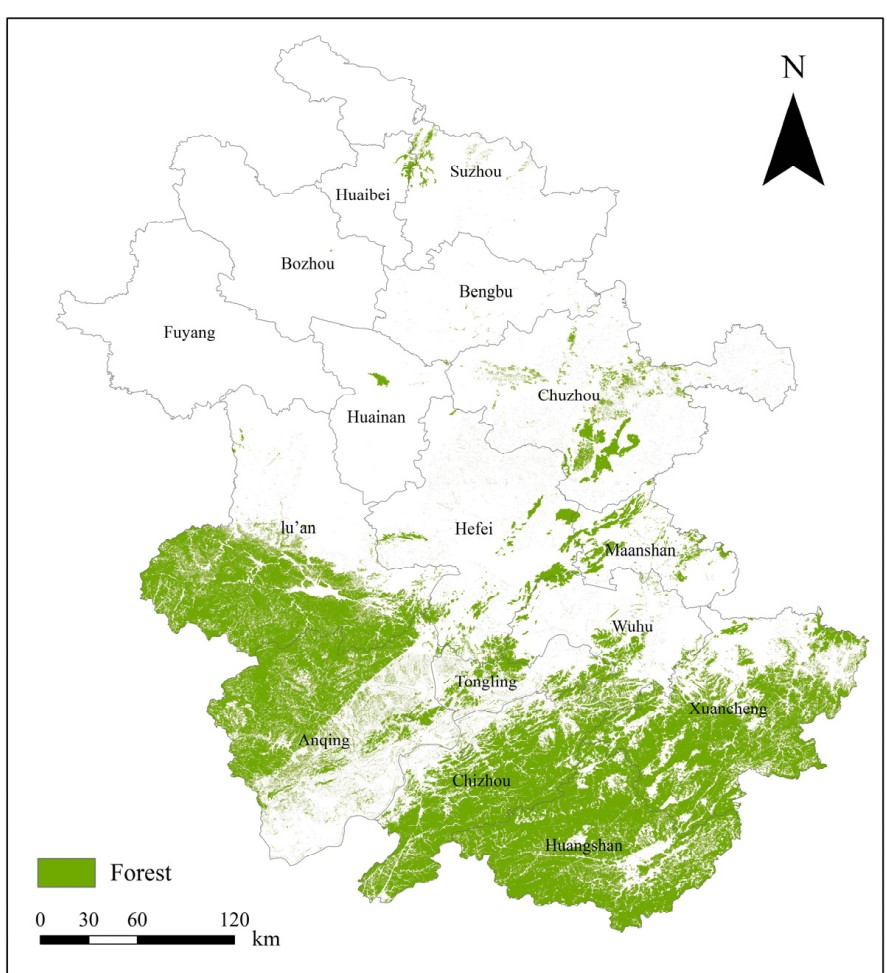

**Figure 3.** GlobeLand30 woodland range map of Anhui Province in 2020.

Based on the forest range in the GlobeLand30 land cover data in 2020, the vegetation sample points were selected on Google Earth Pro and uploaded to the GEE platform. The monthly mean information of the changes in the time-series of vegetation indexes of different forest types in 2020 was extracted based on Landsat 8 images, the forest types in the study area were divided into coniferous forest and other forests. The selected characteristics of sample points are shown in Table 1.

**Table 1.** Interpretation characteristics of forest types from Google Earth Pro high-definition images.

| Forest Types | Example | Description |
|---|---|---|
| Coniferous forest |  | Most of them are distributed in sheets, with uniform tone of dark green and clear boundary with other woodlands |
| Other forests |  | Chaotic shape and uneven tone |

The commonly used vegetation index is mainly composed of the difference and ratio of the reflectance of the red band and the near-infrared band, so as to excavate the important information hidden in vegetation, which can more comprehensively reflect the condition of vegetation coverage and growth. There have been many studies on forest type classification through vegetation index. Considering the objective of this study is to extract coniferous forest areas, and combined with the various characteristics of different vegetation indices, this paper selected five indices, Normalized Difference Vegetation Index (NDVI), Difference Vegetation Index (DVI), Ratio Vegetation Index (RVI), Soil-Adjusted Vegetation Index (SAVI) and Land Surface Water Index (LSWI), to construct the coniferous forest extraction index. See Table 2 for details.

**Table 2.** Vegetation indices for extracting coniferous forest.

| Index | Formula | Reference |
|---|---|---|
| NDVI | $NDVI = \frac{NIR-RED}{NIR+RED}$ | Rouse et al. [22] |
| DVI | $DVI = NIR - RED$ | Demetriades et al. [23] |
| RVI | $RVI = \frac{NIR}{RED}$ | Pearson et al. [24] |
| SAVI | $SAVI = \left(1.1 - \frac{SWIR2}{2.0}\right) * \frac{SWIR1-RED}{SWIR1+RED+0.1}$ | Huete et al. [25] |
| LSWI | $LSWI = \frac{NIR-SWIR1}{NIR+SWIR2}$ | Maki et al. [26] |

Note: NIR, RED, SWIR1 and SWIR2 represent the reflectivity of near-infrared band, red band, shortwave infrared band 1 and shortwave infrared band 2, respectively.

2.4.2. Extraction of Vegetation Index Time-Series Data

Different forest types will show different characteristics as time changes, so the time-series data of vegetation index can be used to identify forest types [27]. On the basis of forest range, the GEE cloud platform was used to calculate the monthly values of NDVI, DVI, RVI, SAVI and LSWI in 2020 for 2000 samples points of two forest types in Anhui Province using expression function.Ee.Reduce.mean( ) function was used to process the mean value of the same vegetation index of the samples in each month, to obtain the monthly mean time-series spectral curves of NDVI, DVI, RVI, SAVI and LSWI of the two forest types in 2020, and to find the best time point for coniferous forest information extraction, so as to build the coniferous forest extraction index and complete the coniferous forest extraction. The index time-series change curves of different forest types are shown in Figure 4.

As shown in Figure 4, the changes curves of each index of coniferous forest and other forests were basically consistent. All the vegetation index values showed a trend of first increasing and then declining. Furthermore, each vegetation index value of coniferous forest was slightly higher than that of other forests. Anhui Province belongs to the transitional region between warm temperate zone and subtropical zone, and is one of the regions with a distinct monsoon climate. From north to south, Anhui Province is divided into four regions: warm temperate zone humid region, north subtropical sub-humid region, north subtropical humid region and middle tropical humid region [28]. With abundant natural

resources, the vegetation in this region is growing well. It can be seen from Figure 4 that each vegetation index shows an overall upward trend from January to April, and the trends of coniferous forest and other forests are basically consistent. From May to August, there is no obvious fluctuation in each vegetation index of coniferous forest, while the vegetation index of other forests witnesses obvious change, forming a trend of first rising and then falling. It can be seen from the analysis of the change curve of vegetation indices that the vegetation indices of coniferous forest and other forests are quite different from May to August. This time-series feature provides a good indicator to distinguish coniferous forest and other forests, which can be used to construct a new vegetation index extracting coniferous forest information.

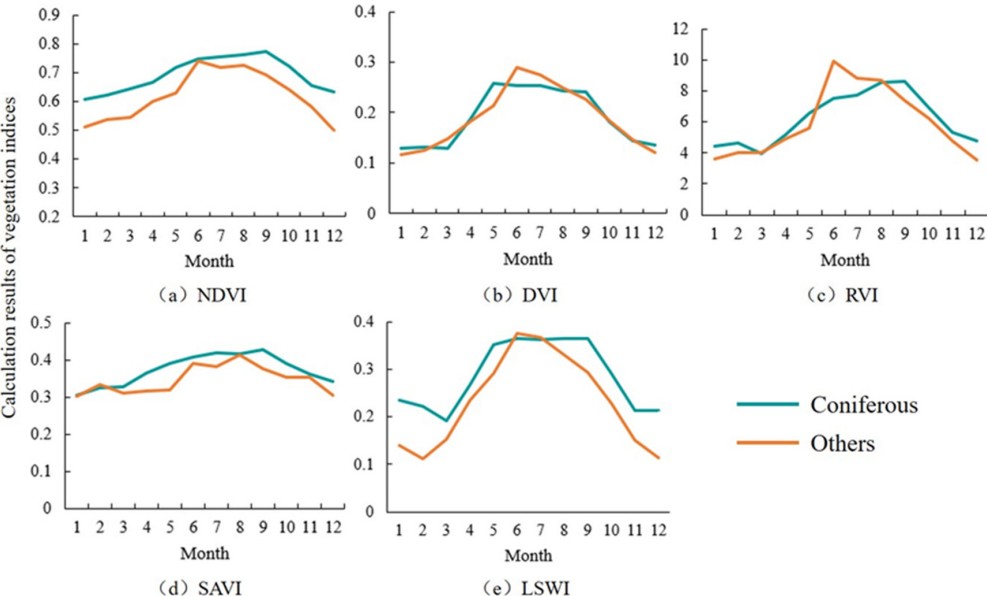

**Figure 4.** Comparison of five vegetation indices time-series change curves of coniferous forest and other forest types.

### 2.4.3. Construction of Normalized Difference Forest Index (NDFI)

According to the results of annual change analysis of different vegetation indices, this study combined different vegetation indices at different times to build a new vegetation index to extract the distribution information of coniferous forests by analyzing the best distinction time of each vegetation index of coniferous forest and other forests. It can be seen from the annual change curve of vegetation index that although the change trend of each vegetation index of coniferous forest and other forest types is nearly the same, there are still some differences, mainly due to the differences in leaf structure characteristics and phenological changes of different forest types [29]. Since NDVI can highlight vegetation information, and facilitate the differentiation of vegetation, this paper used NDVI to participate in construction of the new forest index. At the same time, according to the vegetation index annual change figure (Figure 4) of different forest types, the reflectivity of DVI from May to June, NDVI and SAVI from June to July, RVI from June to August and LSWI from July to August can provide a good differentiation between coniferous forest and other forests. Therefore, the Normalized Difference Forest Index (NDFI) that distinguishes coniferous forest from other forests was constructed by using the time-series change characteristics of each vegetation index. The formula is as follows (Formula (1)):

$$NDFI = \frac{1}{5}\left[\frac{(NDVI_7 - NDVI_6)}{(NDVI_7 + NDVI_6)} + \frac{(DVI_5 - DVI_6)}{(DVI_5 + DVI_6)} + \frac{(RVI_8 - RVI_6)}{(RVI_8 + RVI_6)} + \frac{(SAVI_7 - SAVI_6)}{(SAVI_7 + SAVI_6)} + \frac{(LSWI_8 - LSWI_7)}{(LSWI_8 + LSWI_7)}\right] \quad (1)$$

Among them, NDFI represents the constructed Normalized Difference Forest Index, and $NDVI_6$, $NDVI_7$, $DVI_5$, $DVI_6$, $RVI_6$, $RVI_8$, $SAVI_6$, $SAVI_7$, $LSWI_7$ and $LSWI_8$ represent

the monthly mean value of NDVI, DVI, RVI, SAVI and LSWI, in May, June, July and August, respectively.

In order to better extract the information of coniferous forest, the probability density curve analysis was conducted on NDFI corresponding to different forest types, which was the distribution of all sample points of each forest type in NDFI. The frequency of occurrences of the index values of 2000 samples of two forest types on the NDFI image were analyzed. The horizontal axis represents the change of the NDFI values, and the vertical axis represents the frequency of the sample points (as shown in Figure 5). According to the probability density curve of different forest types, the intersection value of coniferous forest and other forests curve was calculated as the threshold of coniferous forest information extraction. The calculation formula is as follows (Formula (2)):

$$X = (\sigma_1\mu_1 + \sigma_2\mu_2)/(\sigma_1 + \sigma_2) \qquad (2)$$

among which $X$ is the discrimination threshold for extracting coniferous forest information, $\mu_1$, $\mu_2$ and $\sigma_1$, $\sigma_2$ represent the mean value and standard deviation of NDFI of the two types of samples, respectively.

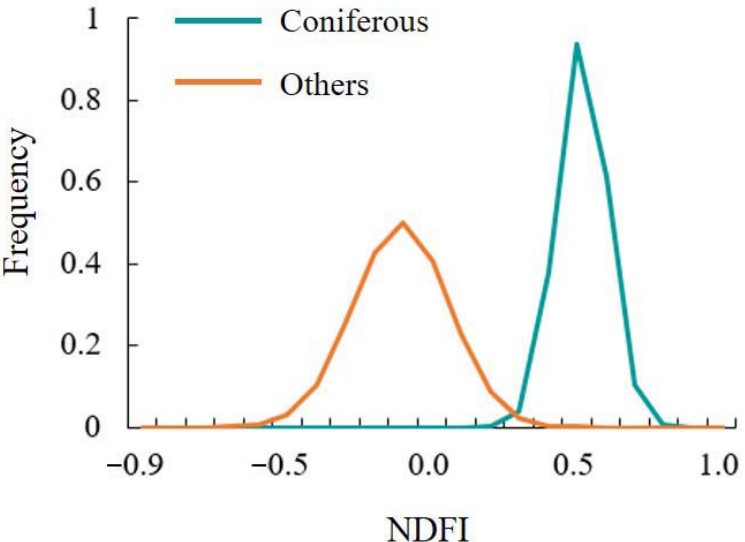

**Figure 5.** Probability density curves of coniferous and other forest type.

As shown in Figure 5, in the NDFI density curve, the best separation threshold value of coniferous forest and other forests can be calculated by using the intersection value calculation Formula (2), where the differentiation point of coniferous forest and other forests is 0.28. So far, the differentiation threshold for extracting coniferous forest information has been determined and can be used for coniferous forest information extraction.

### 2.4.4. Coniferous Forest Information Extraction Based on Time-Series Index Change Analysis

In order to extract coniferous forest information, the forest range in the study area was first extracted using GlobeLand30's land cover data. Then, the forest range was divided into coniferous forest and other forests. After that, the annual change curves of five vegetation indexes were obtained through the GEE platform, so as to obtain the time-series characteristics of vegetation indices to build a new vegetation index NDFI. Then, classification thresholds were extracted by establishing probability density curves of different forest types, from which the extraction of coniferous forest information was completed. The specific steps were as follows: (1) Uploaded the vector data of GlobeLand30's land cover forest range to the GEE cloud platform, and took the forest vector range as the boundary to obtain the Landsat image of Anhui Province in 2020; (2) The GEE cloud platform was used to build a cloud removal algorithm to remove cloud and synthesize the Landsat images

in the forest area of Anhui Province in 2020 on a monthly basis to obtain the monthly cloud removal and synthesis images in 2020. According to the vegetation index formula, monthly images of five vegetation indexes, NDVI, DVI, RVI, SAVI and LSWI, were obtained; (3) Uploaded two forest type sample points to the GEE cloud platform to extract the samples corresponding to different vegetation index values and obtain the annual change curves of five vegetation indexes; (4) NDFI was calculated based on the GEE platform, and NDFI.gt (0.28) was set according to NDFI threshold to complete the extraction of coniferous forest information.

*2.5. Monitoring Model of PWD Based on Landsat Imagery*

2.5.1. Feature Extraction and Selection

Considering the influence of cloud cover on the optical images in the study area and the onset time of the infected host plants (from late August to early November), this paper obtained the cloud removal synthetic image of Anhui Province from August to November in 2021, based on the GEE platform, and extracted the original spectrum of the feature band, vegetation indices, texture information and topographic features from 614 sample points obtained from the field survey in Huoshan County, Anhui Province at the end of October 2021. The characteristic parameters were selected by means of analysis of variance (F test) and importance feature selection, including seven original spectra B1~B7; six vegetation indexes, including Red-Green Index (RGI) [30], Normalized Difference Vegetation Index (NDVI) [31], Normalized Difference Moisture Index (NDMI) [32], Moisture Stress Index (MSI) [33], Normalized Burn Ratio (NBR) [34] and Tasseled Cap Transformation Wetness (TCW) [35]; eight common texture feature metrics, including Mean (MEA), Variance (VAR), Correlation (COR), Contrast (CON), Dissimilarity (DIS), Homogeneity (HOM), Second Moment (SM) and Entropy (ENT); and three topographic features, including Elevation, Slope and Aspect.

Analysis of variance (ANOVA) was usually used to test the significance of differences between the mean values of multiple samples, which can be divided into one-factor analysis of variance (one-way ANOVA) and multi-factor analysis of variance (MANOVA) [36]. This paper use MANOVA as the first step of feature selection. When the significance of the variable is less than 0.05, it means that at the 0.05 significance level, the variable has a significant difference on the occurrence of the dependent variable PWD. This study took the variables with significance less than 0.05 as the next-step monitoring features. Through MANOVA, we obtained 47 features in total, including B2~B4, B6 and B7 in the original spectrum, MSI, NBR, NDMI, NDVI, RGI and TCW8 in the vegetation index, $CON_{65}$, $CON_{75}$, $COR_{65}$, $DIS_{65}$, $ENT_{15}$, $ENT_{75}$, $HOM_{15}$, $HOM_{65}$, $HOM_{75}$, $WEA_{15}$, $WEA_{35}$, $WEA_{45}$, $WEA_{65}$, $WEA_{75}$, $SM_{15}$, $SM_{75}$, $VAR_{15}$, $VAR_{35}$, $VAR_{65}$, $VAR_{75}$, $CON_{69}$, $CON_{79}$, $COR_{29}$, $DIS_{69}$, $DIS_{79}$, $ENT_{79}$, $HOM_{69}$, $HOM_{79}$, $MEA_{69}$, $MEA_{79}$, $SM69$, $SM_{79}$, $VAR_{69}$ and $VAR_{79}$ in the texture information, and Elevation and Aspect in the topographic features.

Through the importance metrics of the random forest method, the classification features with complex relationships can be ranked, and the relative importance of each feature to prediction can be obtained [37]. Generally speaking, for the dataset with relatively large dimension, this method can be used to eliminate the features with relatively small impact in the dataset, and ensure the training speed as well as the accuracy of the data. The two most commonly used methods in ranking important features of random forests are permutation importance and Gini importance. In this paper, based on the results of the variables selected from the analysis of variance, we used R language to process ranking of Gini importance features, and further selected the feature variables. In this study, we set the number of optimal trees (ntree) to 1900, and the number of pre-selected characteristic variables at the optimal tree node (mtry) to 29. The results of the first 10 important characteristics are shown in Table 3.

**Table 3.** Ranking results of Gini importance features.

| Feature | Mean Decrease Accuracy | Mean Decrease Gini |
|---|---|---|
| Elevation | 88.20875 | 38.609919 |
| MSI | 38.38783 | 16.877994 |
| NBR | 36.80615 | 15.824209 |
| B2 | 47.7794 | 14.303737 |
| RGI | 41.53761 | 13.566015 |
| NDMI | 28.44070 | 12.623385 |
| Aspect | 38.16040 | 11.632085 |
| COR$_{29}$ | 30.11671 | 9.689533 |
| COR$_{65}$ | 25.90826 | 9.26472 |
| TCW8 | 29.65953 | 8.234906 |

Based on the ranking results of the above important features, this paper selected the features with an average decreasing Gini value of more than 10, which were the first seven features, Elevation, MSI, NBR, B2, RGI, NDMI and Aspect to construct the classification model of PWD. The infected and non-infected sample points in the field survey were divided into 430 training samples and 184 test samples according to the ratio of 7:3. Based on the GEE cloud platform, the random forest classification method was used to monitor the occurrence of PWD in the epidemic areas of Anhui Province.

2.5.2. Construction of Monitoring Model

The key of constructing the pest monitoring model is to find out the indicators that are sensitive to the infection symptoms of their host plants. At present, there are three types of forest pest monitoring models, namely, various index model, combined model of various channel bands, and mixed model of various channels and ecological factors [38]. In this study, the monitoring model of PWD in the study area was constructed by using the selected characteristic variables and the random forest classification method based on the GEE cloud platform, and the monitoring results were analyzed.

Random forest (RF) is a classifier composed of voting mechanisms of different decision trees. Samples are trained and predicted through multiple decision trees to obtain the final classification results [39]. The random forest classification algorithm is suitable for high-dimensional data processing, and because of the use of random sampling in the classification process, it can reduce the occurrence of the over-fitting phenomenon. At present, this algorithm has been widely used in remote sensing imagery classification, artificial intelligence and other fields. Based on Landsat remote sensing imagery combined with random forest and decision tree algorithms, Huang Fangfang et al. [40] carried out remote sensing monitoring on PWD of *Pinus massoniana* in Yiling District, Yichang City, Hubei Province. The results showed that the random forest classification algorithm with original spectrum and vegetation index had the best classification results, and the classification accuracy reached 80.5%. This research was based on the GEE cloud platform, using ee.Classifier.randomForest( ) function in the random forest classification algorithm, through repeated experiments, selected and set the number of decision trees numberOfTrees in the classifier to 20 at which the classification result was the most stable and the best.

**3. Results**

*3.1. Evaluation of Extraction Accuracy of Coniferous Forest in Anhui Province*

This paper used the method of constructing a confusion matrix to evaluate forest resources classification results in Anhui Province. Using Google Earth Pro software, 1000 coniferous forest sample points and other forests' 1000 sample points were selected within the range of forest resources in Anhui Province as verification samples of classification results. General classification accuracy and kappa coefficient were analyzed and calculated (see Table 4).

**Table 4.** Accuracy evaluation of classification results.

| Forest Type | Coniferous Forest | Other Forests | Total | General Classification Accuracy (%) | Kappa Coefficient |
|---|---|---|---|---|---|
| Coniferous Forest | 870 | 130 | 1000 | | |
| Other Forests | 115 | 885 | 1000 | 87.75 | 0.755 |
| Total | 985 | 1015 | 2000 | | |

As shown in Table 4, the total accuracy of the classification results based on the constructed vegetation index NDFI in Anhui Province in 2020 reaches 87.75%, and the kappa coefficient is 0.755. This shows that the classification results have a high accuracy, and can be used for the extraction of coniferous forest information in this area.

Table 5 shows the coniferous forest area extracted in this paper and the statistical area of sub-compartments taking pine trees as the dominant tree species, as well as the relative error between the two. Figure 6 compares the coniferous forest area extracted in this paper and the distribution area from the sub-compartment survey where the pine trees are the dominant tree species.

**Table 5.** Classification area accuracy evaluation.

| Year | Area of Coniferous Forest Compartment in the Forest Management Inventory (ha) | Area of Coniferous Forest Extracted in This Paper (ha) | Relative Error (%) |
|---|---|---|---|
| 2020 | 1,046,586 | 963,495 | 7.93 |

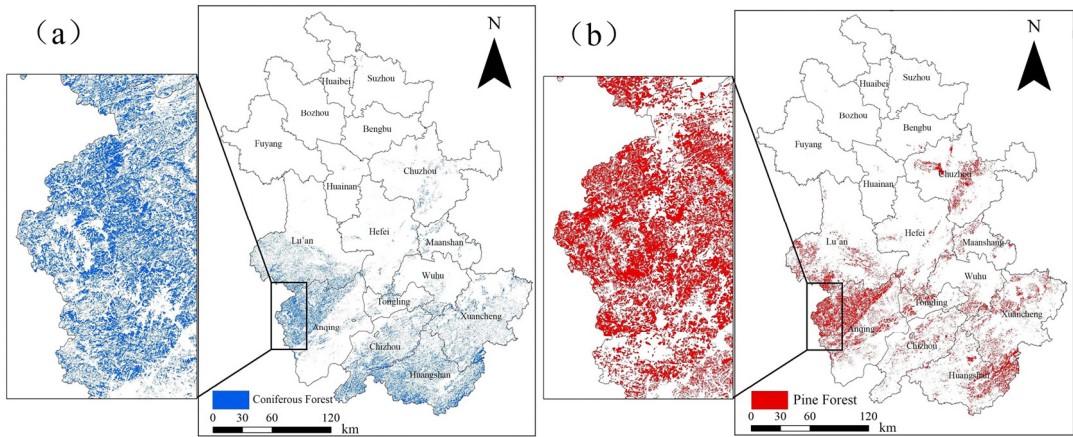

**Figure 6.** Comparison of the extracted coniferous forest area and the distribution area of pine forest sub-compartments. (**a**) Area of coniferous forest extracted in this paper. (**b**) Area of coniferous forest in Forest Management Inventory.

According to the statistical table and the comparison of distribution areas, the extraction results of coniferous forests are basically consistent with the statistical results of sub-compartments of pine forests. The statistical area of the sub-compartments is relatively larger than that of the classification results, and the distribution range is different in some regions, with a relative error of 7.93%. Considering that the sub-compartment area is the forest area with pine trees as the dominant tree species, which is generally larger than the actual pine forest area, this paper believes that the extraction range of the coniferous forest meets the accuracy requirements, and can be used as the monitoring range of the potential PWD occurrence area.

### 3.2. Monitoring of PWD Based on Random Forest

The results show that the overall classification accuracy of random forest is 81.67%, and the kappa coefficient is 0.622. On this basis, we continued to complete the monitoring of PWN epidemic areas in the study area from 2018 to 2021. After the monitoring was completed on the GEE platform, the results were exported to ArcGIS, and the raster data was converted into vector data to calculate the annual disease occurrence area and compare it to the statistical data of the disease occurrence area over the four years. The relative error evaluation index was used to complete the precision evaluation of the disease monitoring results over the years. See Table 6 for accuracy evaluation of the monitoring area over the years and Figure 7 for classification of monitoring results.

**Table 6.** Accuracy evaluation of monitoring area over the years.

| Year | Statistical Area (ha) | Monitoring Area (ha) | Relative Error (%) |
|------|----------------------|----------------------|--------------------|
| 2018 | 20,820 | 26,871 | 29.06 |
| 2019 | 110,000 | 137,772 | 25.24 |
| 2020 | 101,333 | 128,142 | 26.45 |
| 2021 | 92,700 | 115,274 | 24.35 |

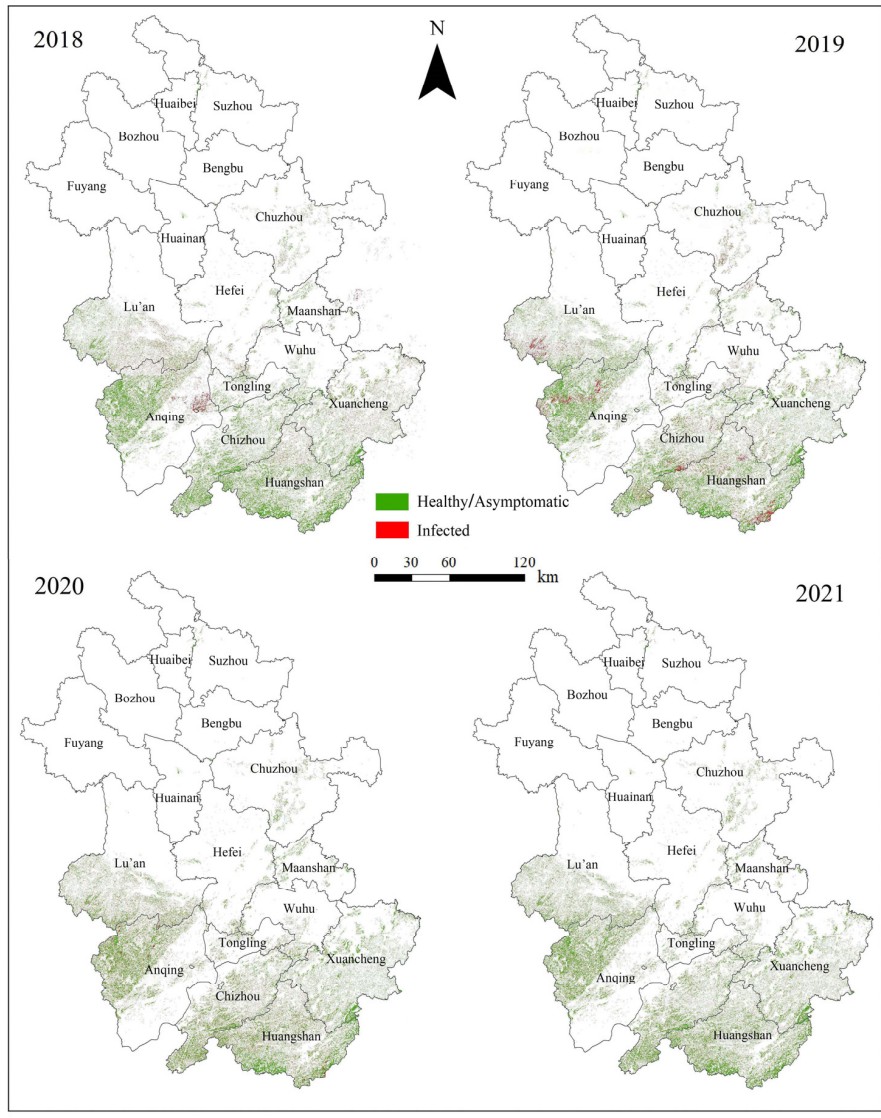

**Figure 7.** Distribution map of PWD from 2018 to 2021.

As shown in Table 6, the relative errors in 2018, 2019, 2020 and 2021 are 29.06%, 25.24%, 26.45% and 24.35%, respectively, and the precision is higher than 70%, which can be used as the basis for focal monitoring of PWD epidemic areas. The monitoring areas are larger than the statistical areas over the four years, which may be due to the fact that the monitoring data source of this paper uses Landsat images which have a lower spatial resolution of 30m. In addition, during the monitoring process, this study sticks to the rule that if there is infected wood in the pixel, the entire pixel is the infected area, so the monitoring results are larger than the statistical results.

In 2017, China began to implement the forest leader system reform. Anhui Province gradually established a complete forest leader system, ensuring the optimal allocation of resources in terms of manpower, technology and equipment, which plays a crucial role in the prevention and control of PWD. According to the monitoring results of PWD in Anhui Province as shown in Figure 7, from 2018 to 2021, the disease occurrence areas showed a downward trend as a whole, and the distribution of epidemic areas gradually changed from local aggregation to discrete type. In September 2018, the Anhui Provincial Government launched a special campaign to control PWD, fully implementing the prevention and control idea of taking clearing and cutting infected trees as the core and managing the source of infected trees as the root. Due to the timeliness of the achievements of the special campaign to kill PWN, the area and range of the disease in 2018 were the smallest, mainly in the middle and southeast of Anhui Province. However, because the host plant infected by PWN first changes its internal physiological parameters, with the external symptoms having the feature of time delaying, it will not be thorough to only rely on human visual identification and cleaning of infected trees in a short period of time. In 2019, the disease was still relatively serious, and the disease occurrence areas expanded from the middle to the west and south, respectively, with local aggregation distribution. With the gradual improvement of PWD prevention and control measures, although the range of disease occurrence areas expanded in 2020, the characteristics of disease occurrence areas changed from clustered distribution to discrete points distribution. In 2021, the disease occurrence areas continued to show a decreasing trend.

*3.3. Disease Degree Analysis*

In order to study the degree of disease, this paper, based on the monitoring results, classified the degree of disease according to the infected area of PWN in the grid and the infection rate obtained from the total area of coniferous forest. Before disease degree division, the grids were constructed in ArcGIS based on coniferous forest. Considering the size of the study area and other factors, the grid size was set to 10 km $\times$ 10 km. Then, the vectorized classification results of each year were imported into ArcGIS for classification post-processing. We calculated the infected area of each spot and removed the small spots caused by the salt-and-pepper phenomenon based on pixel classification, then merged the adjacent spots. Then we overlaid the grids and the classified spots of each year, and counted the ratio of the area of infected spots to the area of coniferous forests in each grid, then completed the disease degree division according to the ratio (as shown in Figure 8).

According to statistics, the range of infection rates of PWD in 2018, 2019, 2020 and 2021 were 0~67%, 0~59%, 0~47% and 0~35%, respectively. In this paper, they were divided into four categories: healthy/asymptomatic, light, moderate and severe. According to the grading map of PWD degree, in 2018, there were a small number of diseases with severe degree in the middle of Anhui Province, while healthy/asymptomatic and light degree accounted for the majority. In 2019, except for a small number of moderate diseases in the southwest and south part of Anhui Province, other regions showed healthy/asymptomatic and light distribution. In 2020, the disease grade was healthy/asymptomatic and mild, mainly distributed in the southwest of Anhui Province. In 2021, the health/asymptomatic degree was the main level, indicating that with the provincial government's efforts to prevent and control PWD, great achievements have been made.

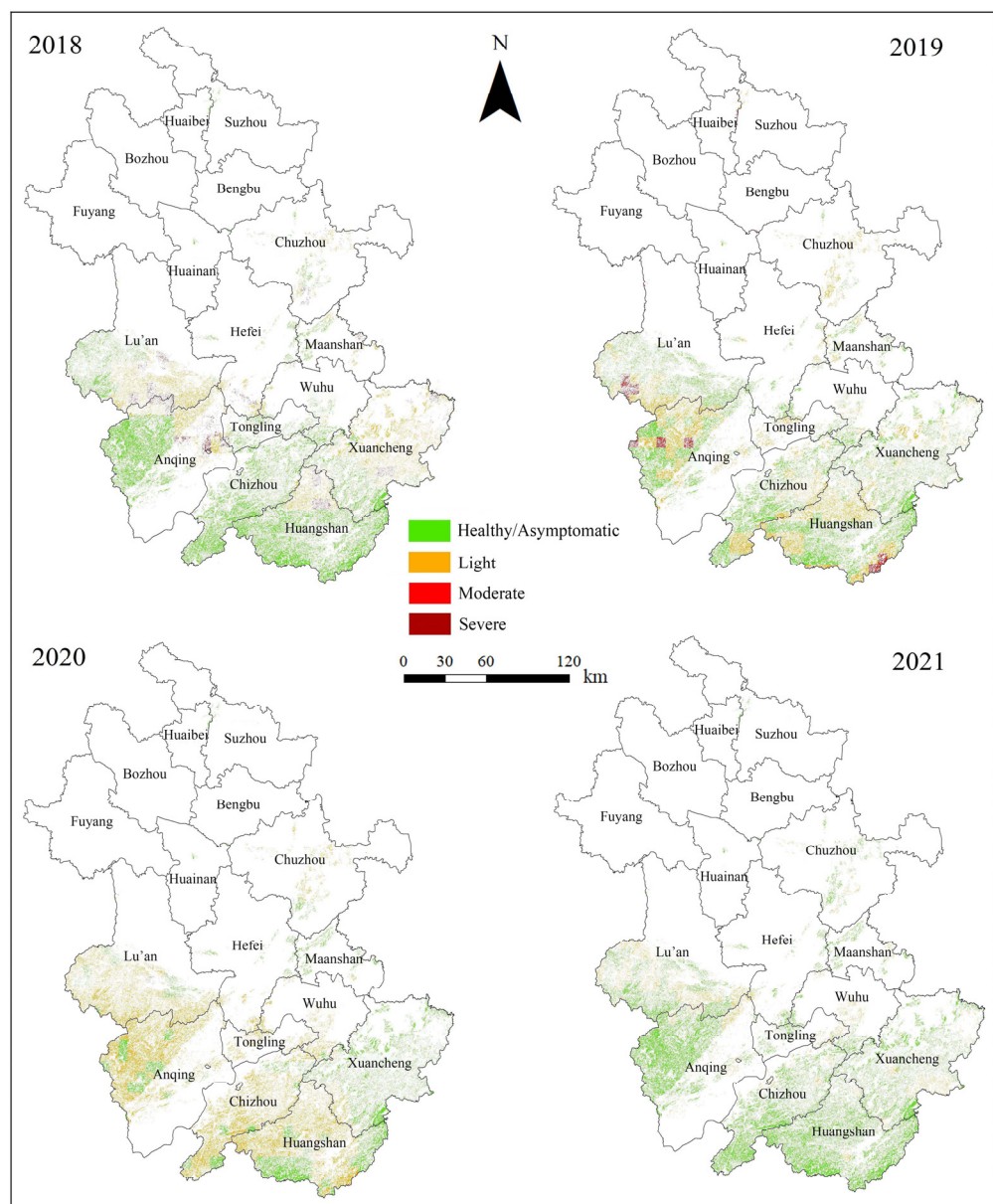

**Figure 8.** Grading map of PWD degree from 2018 to 2021.

## 4. Discussion

### 4.1. Effectiveness and Feasibility of NDFI for Extracting Coniferous Forest Distributions

PWN is a destructive disease of pine trees, and its rapid propagation is an important cause of death of pine trees [41]. With the increasing potential adaptability of PWN in China in recent years, the host plants of PWN have gradually expanded from dozens of pine forest trees such as *Pinus massoniana* and *Pinus densiflora* to other non-Pinus coniferous trees [42]. In this study, the forest types in the GlobeLand30 land cover were taken as the forest range of the study area. Based on the GEE cloud platform, through the study of the time-series characteristics of the annual change curve of five vegetation indexes, NDVI, DVI, RVI, SAVI and LSWI, we found that the reflectance values of different vegetation indexes of coniferous forests and other forests were significantly different from May to August. Using this feature, a new vegetation index NDFI was constructed based on different vegetation indexes. By means of probability density curve analysis, the differentiation threshold of NDFI is 0.28, which is used for information extraction of the host plant range. The results show that the overall accuracy of coniferous forest information extraction results reaches 88.75%, kappa coefficient is 0.755, showing the extraction method is quite

effective. The method of constructing a new vegetation index has a certain portability. Liu et al. [43] established hyperspectral vegetation index and differentiation equation of wheat dry-hot wind damage level index according to the spectral range of EOS/MODIS visible light channel by constructing RVI, NDVI, PVI, DVI and other vegetation indexes. The results showed that the reflectance spectra of spring wheat with different degrees of dry-hot wind damage were significantly different, which provided a basis for remote sensing monitoring and assessment of spring wheat dry-hot wind damage. Zhang et al. [44] combined the mechanism of nitrogen movement between different functional leaves of rice, and built a red edge curve shoulder angle vegetation index (RSAVI) based on the red edge position and the red edge slope to monitor the nitrogen nutrition status of rice. He analyzed the correlation between nitrogen content and RSAVI at different growth stages. The results showed that RSAVI was significantly correlated with leaf nitrogen content, and the correlation coefficient was between 0.867 and 0.938, the models all passed the 0.01 level test. The study shows that it is feasible to use RSAVI to estimate rice nitrogen nutrition. According to the information extraction results of the coniferous forest and the statistical data of the area and distribution range of the pine forest, the extraction results of the coniferous forest are basically consistent with the statistical distribution range of the sub-compartments of the pine forest, and the relative error of the area is 7.93%, which is completely available in PWD monitoring practice at province level.

### 4.2. Practicability and Popularization of Periodic Monitoring of PWD

Global climate change has a tremendous impact on all aspects of society and nature [45], PWD has caused great damage to the pine forest resources in China, so timely and effective control of the spread of PWD has become the working focus of governments at all levels. With the development and application of remote sensing technology and artificial intelligence, intelligent remote sensing monitoring with remote sensing technology as the platform and artificial intelligence technology as the core has become a feasible and high-precision automatic monitoring method for PWD. In view of the widespread range of PWN and the high requirements for monitoring accuracy, the multi-platform remote sensing monitoring method of PWD with satellite wide area survey, UAV regional detailed survey and artificial ground verification as the main line has become one of the important research methods of PWD control [46]. In this study, various classification features were extracted through GEE cloud platform, and seven classification features including Elevation, MSI, NBR, B2, RGI, NDMI and Aspect were selected after feature evaluation by using MANOVA and importance feature ranking. The monitoring model of PWD was established using random forest classification algorithm based on the extracted coniferous forest range. The overall classification accuracy is 81.67%, and the kappa coefficient is 0.622. Analyzing the temporal and spatial variation characteristics of PWD degree in the study area by grid division, the study found that from 2018 to 2021, the occurrence area range of PWD gradually changed from aggregated distribution to discrete distribution. The overall disease degree gradually decreased, while the proportion of healthy/asymptomatic and mildly infected areas gradually increased, showing that the government has made great achievements in coordinating the prevention and control of PWD. The value and significance of this study is that medium resolution satellite remote sensing can achieve regional monitoring periodically at low cost. Although it cannot achieve the high accuracy of UAV monitoring, it can find suspected disease areas in a large range through repeated observation data in a short period, providing a basis for airborne fine monitoring, which is practical and promotional.

### 5. Conclusions

This paper took Anhui Province as the study area, a large area of which was suffering from PWD. We used Landsat continuous multi-year vegetation index data to establish a new vegetation index NDFI to extract coniferous forest information in Anhui Province for its disease prevention and control. With the support of a big data processing platform

and long time-series vegetation index remote sensing data in field investigation, spatial analysis technology and machine learning technology, we realized the identification of PWN host plants and the construction of disease monitoring model, and analyzed the characteristics of spatial and temporal changes of PWD level in the study area. It provides a feasible application scheme for the monitoring, prevention and control of PWD using satellite-borne remote sensing technology.

In recent years, many scholars have conducted a lot of research on the characteristics, mechanism and dynamic spread of PWN, and made great progress in theory and practice, which has played an important role in providing decision-making support for government departments. There is no denying that the spread of PWD is a complex problem of interaction between human beings and the environment, and its scientific research still faces many difficulties and challenges. For example, for a long time, early recognition of PWD from remote sensing images, 'early' definition and real-time dynamic spread monitoring of PWD have been the expected goals of scholars, but the effect of remote sensing information recognition is not very ideal at present. The realization of fine early identification of PWD and real-time dynamic spread monitoring can greatly improve the accuracy and efficiency of monitoring PWD, and will provide more powerful support for national economic construction. Therefore, comprehensive research based on multi-scale and high-precision remote sensing big data will be an essential direction in the study of spatial patterns of PWN in the future.

**Author Contributions:** Conceptualization, L.L. and Y.C.; methodology, L.L. and X.Z.; software, L.L. and Y.C.; validation, L.L.; formal analysis, L.L.; investigation, L.L., Y.C. and X.J.; resources, Y.L. and G.L.; data curation, L.L. and Y.C.; writing—original draft preparation, L.L. and Y.C.; writing—review and editing, L.L., S.S. and X.Z.; visualization, L.L.; supervision, X.Z.; project administration, X.Z.; funding acquisition, X.Z. All authors have read and agreed to the published version of the manuscript.

**Funding:** This research was funded by the National Natural Science Foundation of China (31870534) and China EU Science and Technology Cooperation Phase V (59257).

**Institutional Review Board Statement:** The study did not require ethical approval.

**Informed Consent Statement:** Not applicable. The study not involving humans.

**Data Availability Statement:** The data is not available because the team data involves privacy issues.

**Acknowledgments:** The authors would like to thank Mengyu Chen for his assistance in the field investigation.

**Conflicts of Interest:** The authors declare no conflict of interest.

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
