# Peer review of "Remote Sensing Monitoring of Pine Wilt Disease Based on Time-Series Remote Sensing Index"

_remotesensing, doi:10.3390/rs15020360_

Round 1
Reviewer 1 Report
I reviewed the manuscript remotesensing-2116712 entitled « Remote Sensing Monitoring of Pine Wilt Disease Based on Time-series Remote Sensing Index». This paper aims at proposing a new index NDFI for extracting the coniferous forest, and using random forest algorithm to monitor PWD based on Landsat time-series images. Then, the authors has analyzed the spatio-temporal dynamics of PWD. The topic of this manuscript has a great significance for monitoring and preventing PWD spread in Anhui province quickly and accurately. This paper has a complete structure and sufficient arguments and results, however, there are several aspects should be considered before potential publication from my perspective:
1. I think the figure 2 does not suitable for description in section 2.3. The technical route in figure 2 contains the entire content of the article, including section 2.3 and section 2.4, so I think the author should add a section before section 2.3 as “method”, to describe the monitoring method of this paper. And then, the section 2.3 and section 2.4 is the next part.
2. The figures in this manuscripts should be promoted, the size, format, fonts, etc. should accord with the requirements of the journal, epically Figure 3, figure 7, figure 8.
3. In section 2.3.1, the forest types were divided into coniferous forest and other forests, and the vegetation sample points were selected based on the forest range in Global Land 30. From I known, the GL30 data only contained forest type, which did not distinguish between coniferous and broadleaved forests. So I confused that whether other auxiliary data is used to divide the vegetation sample points. I suggest that the author should revise second paragraph to describe more specific of the train samples in section 2.3.1(Line 212-217).
4. In section 2.3.2, the author has selected the mean() to generate the monthly data of each indexes, I want to know how can the author ensure monthly coverage of landsat images in the study area. If there were deletions of images, how does the authors deal with this problem?
5. Line 238, sample should be revised as samples, I suggest that the author should check the grammar of this paper.
6. Line 344-345, ntree and mtry should be added a parenthesis as variables of random forest model.
7. Section 3.1 is the specific steps, which is not suitable in the results, so I suggest that the author should modify the structure of this paper, the author can modify the structure with comments 1.
8. Line 405, “It can be seen from table 4 that the……” could be removed, the language of research paper should be concise.
9. Line 433-Line437, this part should be listed in materials and methods (Section 2). In section 3 –Results, the author should only presented the results of each steps, same questions in Line 481-486.
10. In figure 7, the distribution of PWD is highly significant in 2017 and 2018, but I noticed the infected area is quite fuzzy, the author could add the contrast of different types or choose another expression.
11. Line 496-497, “It can be seen from….” should be revised.
In the present form, firstly, this research paper is meaningful, which has proposed novelty a vegetation index NDFI to extract coniferous forest used Landsat continuous multi-year vegetation index data, it may be of interest to other researcher. But secondly, this manuscript should be promoted with careful revision. There are more methodology, and data acquisition in the results.
In short, the paper could be structured much better with a major revision is made to address these shortcomings.

Reviewer 2 Report
After reviewing the article with title: Remote Sensing Monitoring of Pine Wilt Disease Based on Time-series Remote Sensing Index, I conclude that it is an interesting article, dealing with an up-to date subject. Moreover, it is well- organized and uses new technologies to achieve the main goals. Some issues need to be revised and correct from the others are listed below:
Line 44. The insect outbreak is also referred as the COVID virus of pinewoods.
Line 77. Recent studies show that Landsat time series efficiently capture the extent of forest disturbances and are reliable for muli-temporal ecosystem services change due to vegetation damages (https://doi.org/10.1016/j.catena.2022.106564) as well as UAV high resolution spectral images (https://doi.org/10.1016/j.jag.2022.102947).
Line 167. How the number of sample size was estimated? The 614 trees linked with what significant level?
Do you downscale the resolution based on panchromatic band?
Figure 2. The processes performed using google earth engine could be filled with a different color in flowchart.
The advantages of using google earth engine rather than classical approaches is missing in the text.
Line 219. Some references about the use of the proposed index for similar purposes. Also, a well-known index appropriate for the article scope and easily extract from Landsat 8 through GEE is the Vegetation health index (VHI). Consider its possible use.
Line 280. The construction of an index using PCA from the already estimated vegetation index could be an alternative.
Line 496. How the class borders were determined (healthy/asymptomatic, light, moderate and severe).
Line 568. Add some direction for future research.
Round 2
Reviewer 1 Report
The manuscript has been revised as first comments, the quality and structure of this manuscript also have been improved to meet the requirements of journal publication.
I have no more question about this manuscripts.
Author Response
Dear reviewers,thank you again for taking the time to review this manuscript!We would like also to thank you for allowing us to resubmit are revised copy of the manuscript.
Thanks again!
Finally, on behalf of all the authors, I would like to thank the reviewers again for their valuable suggestions. I wish you good health, happy life and smooth work!
Reviewer 2 Report
The article is not adequately referenced and few references provided. Moreover, the suggestions for some reference about landast imageries time series analysis to capture landcover changes ignored in the preivious review round.
